# Secure and Privacy-Preserving Intrusion Detection and Prevention in the Internet of Unmanned Aerial Vehicles

**DOI:** 10.3390/s23198077

**Published:** 2023-09-25

**Authors:** Ernest Ntizikira, Wang Lei, Fahad Alblehai, Kiran Saleem, Muhammad Ali Lodhi

**Affiliations:** 1School of Software, Dalian University of Technology, Dalian 116024, China; ntiernest@gmail.com (E.N.); kiransaleem13@hotmail.com (K.S.); alilodhi30@gmail.com (M.A.L.); 2Department of Computer Science, Community College, King Saud University, Riyadh 11437, Saudi Arabia

**Keywords:** unmanned aerial vehicles (UAVs), intrusion detection, privacy preserving, Internet of UAVs, federated learning, real-time decision mechanism

## Abstract

In smart cities, unmanned aerial vehicles (UAVS) play a vital role in surveillance, monitoring, and data collection. However, the widespread integration of UAVs brings forth a pressing concern: security and privacy vulnerabilities. This study introduces the SP-IoUAV (Secure and Privacy Preserving Intrusion Detection and Prevention for UAVS) model, tailored specifically for the Internet of UAVs ecosystem. The challenge lies in safeguarding UAV operations and ensuring data confidentiality. Our model employs cutting-edge techniques, including federated learning, differential privacy, and secure multi-party computation. These fortify data confidentiality and enhance intrusion detection accuracy. Central to our approach is the integration of deep neural networks (DNNs) like the convolutional neural network-long short-term memory (CNN-LSTM) network, enabling real-time anomaly detection and precise threat identification. This empowers UAVs to make immediate decisions in dynamic environments. To proactively counteract security breaches, we have implemented a real-time decision mechanism triggering alerts and initiating automatic blacklisting. Furthermore, multi-factor authentication (MFA) strengthens access security for the intrusion detection system (IDS) database. The SP-IoUAV model not only establishes a comprehensive machine framework for safeguarding UAV operations but also advocates for secure and privacy-preserving machine learning in UAVS. Our model’s effectiveness is validated using the CIC-IDS2017 dataset, and the comparative analysis showcases its superiority over previous approaches like FCL-SBL, RF-RSCV, and RBFNNs, boasting exceptional levels of accuracy (99.98%), precision (99.93%), recall (99.92%), and *F*-Score (99.92%).

## 1. Introduction

Recently, smart cities have gained significant traction as urban areas embrace advanced technologies to improve efficiency, sustainability, and quality of life [1,2]. To integrate diverse urban systems and improve the quality of services provided to residents, smart cities incorporate cutting-edge technologies such as the Internet of Things (IoT), big data analytics, and artificial intelligence [3,4,5,6].

UAVs offer unique advantages in smart-city environments, making them essential for a wide variety of programs. These flights offer actual-time surveillance and monitoring capabilities, permitting authorities to accumulate essential statistics from quite a few resources and locations wherein UAVs have verified benefits in range of scenarios from site visitor tracking to environmental evaluation and disaster response for public protection [7,8]. The ability to navigate challenging terrain and far off areas with ease makes them priceless assets for growing situational focus and a rapid response time [9,10,11].

### 1.1. Problem Statement and Motivation

As UAVs become increasingly prevalent in smart cities, safety and privacy concerns have risen accordingly [12]. The use of UAVs raises the issues of safety breaches and unauthorized access to sensitive information. Since UAVs gather and transmit a tremendous amount of data, privacy concerns and intrusion detection and prevention become crucial. The widespread integration of UAVs in smart cities has introduced a new dimension to urban management and public safety. These versatile aircraft offer real-time data collection capabilities across various domains, from traffic monitoring to disaster response. However, with this proliferation comes a critical challenge: ensuring the security and privacy of collected and transmitted data [13,14].

One of the primary concerns is the vulnerability of UAV networks to intrusions, which can have severe repercussions on public safety and data integrity. As UAVs become indispensable for critical applications, safeguarding their operations against potential threats becomes paramount. Additionally, the sensitive nature of the data they handle necessitates robust privacy-preserving mechanisms [15,16].

Existing security measures often fall short in addressing the unique challenges posed by UAV deployments in smart cities. Conventional intrusion detection systems and privacy protection techniques are not tailored to the dynamic and resource-constrained nature of UAV networks [17,18]. This gap in existing solutions underscores the need for a specialized framework designed specifically for this context. Additionally, the current landscape of UAV security and privacy solutions lacks a comprehensive and tailored approach for smart cities. Many existing approaches focus on conventional network security measures, overlooking the intricacies of UAV operations [19,20]. Furthermore, privacy-preserving techniques often do not account for the dynamic nature of UAV networks, leading to suboptimal protection [21,22].

This apparent gap in the existing literature prompted the development of the SP-IoUAV model. Our aim was to fill this gap by developing an innovative framework that not only addresses the security and privacy concerns specific to UAV deployments in smart cities but also employs cutting-edge techniques to ensure data integrity and confidentiality.

### 1.2. Proposed Work Contributions

The SP-IoUAV model makes the following contributions:**Innovative Privacy-Preserving Mechanism:** Our novel privacy-preserving mechanism combines federated learning, differential privacy, and secure multi-party computation in a powerful way to preserve privacy. This mechanism significantly enhances the detection and prevention of intrusions in UAV networks while ensuring the comprehensive protection of sensitive data.**Robust Intrusion Detection Engine:** An intrusion detection engine built on a CNN-LSTM deep neural network forms the basis of our model, which is robust in nature. Using this engine, UAVs can detect anomalies and classify them in real time, allowing them to identify security threats quickly.**Proactive Real-Time Decision Mechanism:** In addition to providing instant alerts and automatic blacklisting, our model also features a cutting-edge real-time decision mechanism and automatic blacklisting. In order to prevent potential breaches, this dynamic system responds quickly, notifies relevant personnel, and takes decisive action when intrusions are detected.**Multi-factor Authentication for Enhanced Security:** For enhanced security, we use multi-factor authentication (MFA) to control database access to the intrusion detection system (IDS). In addition to providing an additional layer of protection against unauthorized access, MFA demonstrates the robustness of our overall system and reassures users of its integrity.

### 1.3. Paper Organization

The rest of this paper is organized as follows: Section 2 reviews the related works in the literature. Section 3 details the methodology, algorithms, and design of our proposed model. The Experimental Findings and Analysis section is presented in Section 4, divided into four subsections covering system requirements, dataset description, simulation setup, and comparative analysis. Lastly, in Section 5, we conclude this research work and outline future research directions.

## 2. Related Works

The Literature Survey section reviews existing research on secure and privacy-preserving intrusion detection for UAVs, providing insights for our proposed SP-IoUAV model.

The authors of [17] introduced a hybrid ML approach, combining logistic regression and random forest, to classify data instances for enhanced privacy and security protection in drones. The technique demonstrated a remarkable accuracy of 98.58%. However, to ensure proper performance evaluation, the authors did not manage to clearly illustrate how their methodology can provide privacy and security protection in UAVs.

The authors of [18] conducted an investigation into security challenges within UAV networks, specifically concerning eavesdropping attacks enabled by the broadcast nature of wireless channels and wide aerial coverage. They explored the application of machine learning techniques to decrypt encrypted locations derived from wireless data transmission. To counter machine-learning-based attacks, they proposed a location protection approach utilizing random linear network coding and randomized encryption keys. The research demonstrated the neural network’s capability to successfully decrypt encrypted locations using conventional protection methods. However, this highlights the necessity for implementing stronger security measures to safeguard sensitive location data transmitted in UAV networks over wireless channels.

The authors of [23] proposed two frameworks, one for creating test data features from wireless signals and another for generating training data to detect eavesdropping attacks in UAV-aided wireless systems. Their results showed that OC-SVM outperformed k-means in terms of stability, while k-means clustering performed better when the eavesdropper used high transmission power. However, the process of detection remained unclear.

The authors of [24] introduced a supervised machine learning approach utilizing an artificial neural network to detect GPS spoofing signals. Various features, including pseudo-range, Doppler shift, and signal-to-noise ratio (SNR), were employed to classify GPS signals for detecting GPS spoofing attacks on unmanned aerial systems. The results demonstrated the high probability of detection and low probability of false alarms achieved by their proposed method. However, the details of the methodology remain unclear.

The authors of [25] introduced a long short-term memory (LSTM) recurrent neural network method for UAV anomaly detection. Initially, a prediction model was formulated based on a training dataset containing normal data only, enabling the prediction of data at a subsequent time. Subsequently, by considering the prediction results during the training phase, an estimation of the prediction uncertainty was obtained. Finally, anomaly detection was accomplished by comparing the prediction value with the uncertain interval. The proposed method was validated using real UAV sensor data with point anomalies in north velocity and pneumatic lifting velocity, showing its effective ability to detect point anomalies. However, using a single neural network method may not be as effective as using a hybrid architecture, which could potentially offer improved performance in UAV anomaly detection.

The authors of [26] proposed an approach that addresses data privacy in UAVs using blockchain technology. They analyzed security solutions combining machine learning and blockchain for UAV collaborative applications. The findings show that machine learning enhances UAV security, and blockchain offers decentralized security. The hybrid model of machine learning and blockchain improves data reliability. This approach can benefit diverse applications, including healthcare and industries where security and data quality are vital. However, further research is needed to explore the full potential of blockchain technology for decentralized UAV security.

The authors of [27] proposed a powerful DL-based blockchain IDS named BIIR for secure IoD environments. This system utilizes RL and RBFNN as agents, representing the DL component of the technique. It achieved high accuracy in identifying attacks even with an increased number of drones, outperforming state-of-the-art techniques. The BIIR approach demonstrated resilience against various known threats in real-world IoD application scenarios. However, it is important to note that our proposed SP-IoUAV model complements the BIIR approach by addressing additional security and privacy challenges in the Internet of Unmanned Aerial Vehicles.

The authors of [28] proposed an advanced IDS for UAV swarms to detect in-flight anomalies and network attacks, addressing security challenges in UAV technology. The IDS offers wider attack coverage, privacy protection, and hardware independence. However, there is a potential gap in evaluating the system’s real efficiency and impact in practical scenarios on hardware components. Additionally, further exploration of using blockchain for enhanced privacy and testing the IDS on a swarm of UAVs could be considered.

The authors of [29] introduced a collaborative intrusion detection algorithm, utilizing CGAN-LSTM with blockchain-empowered distributed federation learning. This algorithm achieves exceptional accuracy exceeding 95% and outperforms other methods. However, to achieve even better performance, it is advisable to explore an ensemble of secure and privacy-preserving approaches.

The authors of [30] presented an intelligent intrusion detection framework empowered by mobile edge computing technology for detecting and predicting diverse attacks in the UAV network. The proposed optimized random forest model, integrated into dedicated UAV-MEC servers, demonstrated efficient attack detection in various UAV network zones. However, since the framework relies on a single DNN algorithm, a combination of multiple secure and privacy-preserving approaches could potentially improve the accuracy of attack detection even further.

The authors of [31] proposed a novel IDS framework based on federated continuous learning with stacked BLS learning systems. They employ DTN to enhance attack detection by decentralizing the learning process. The framework addresses data silos, ensures privacy and security, and achieves high accuracy and training efficiency. However, combining this approach with other secure and privacy-preserving techniques could further enhance performance.

The authors of [32] introduced an optimized hierarchical anomaly-based intrusion detection system specialized in identifying and alerting lethal attacks in military operations within Internet of Drones networks. By utilizing an optimized hyperparameters algorithm and randomized search cross-validation, an efficient random forest classifier was designed as the baseline algorithm for the IDS. The simulation results confirmed the superiority of the proposed model, which outperformed selected optimized models based on essential performance metrics. However, the reliance on a single classifier could be further improved by exploring a combination of secure and privacy-preserving approaches for enhanced attack detection accuracy.

The authors of [33] proposed a supervised ML approach for detecting Sybil attacks in FANETs-based IoFT. While achieving a high classification accuracy of over 91%, the study lacks clarity in the analysis and does not fully demonstrate the model’s potential in handling diverse attacks in UAV networks. A more comprehensive evaluation of the proposed model under varied attack scenarios is needed for real-world applicability. Likewise, FANETs have been utilized in UAV networks to optimize communication using the fisheye state routing (FSR) protocol [34].

The authors of [35] proposed an innovative solution focused on preserving the privacy of lane images. Their approach combines ELA, ANN, and a U-Net model for real-time lane detection. The final step employs proxy re-encryption with RSA and ECC algorithms to ensure image integrity. However, the approach primarily addresses image security in rural lane contexts. In contrast, our research takes a more comprehensive approach, encompassing security and privacy concerns across the entire UAV ecosystem. This includes a significant emphasis on robust intrusion detection mechanisms, crucial for safeguarding operations against a range of potential threats.

In the literature, many authors have provided different methods for intrusion detection and prevention in the Internet of UAVs. However, it is crucial to acknowledge that this progression is not without its challenges. Table 1 illustrates the methods, limitations, and advantages of each research work cited in the literature survey.

One significant limitation in the existing literature revolves around the security and privacy concerns associated with UAV operations. As UAVs become increasingly integrated into urban environments, safeguarding data transmission and preventing unauthorized access have emerged as critical issues. In response to these challenges, our proposed SP-IoUAV model leverages a novel privacy-preserving mechanism, combining federated learning, differential privacy, and secure multi-party computation. This approach not only enhances the detection and prevention of sensitive data breaches but also addresses the broader spectrum of security and privacy concerns within the UAV ecosystem. By mitigating these limitations, our model seeks to provide a robust framework for secure and privacy-preserving UAV operations within smart cities.

## 3. The Proposed SP-IoUAV Model

We present in this section a model for detecting and preventing intrusions with security and privacy in the Internet of UAVs. In order to ensure data privacy during data transmission, the model uses advanced techniques such as federated learning, differential privacy, and secure multi-party computation. As our IDS architecture enhances security and privacy for UAVs by adding multi-factor authentication (MFA) to secure database access, we are facilitating seamless collaboration and efficient decision making within the smart-city environment. The proposed model is depicted in Figure 1, while in Table 2, we describe and define all symbols, variables, and notations used in this research work.

### 3.1. Unmanned Aerial Vehicles (UAVs)

Using UAVs as its backbone, the proposed model collects information and disseminates it in the IoUV ecosystem of unmanned aerial vehicles. In order to collect real-time data from their surroundings, such as video feeds, environmental parameters, and network traffic, these self-driving vehicles are equipped with sensors and communication modules. In the proposed model, UAVs play the following roles:**Data Collection:** UAVs continuously gather visual and non-visual data from their environment, including images, videos, environmental parameters, and network traffic data.**Data Tagging:** UAVs add unique identifiers or tags to each data point before transmission to indicate their source, maintaining privacy during data sharing.**Data Encryption:** UAVs employ encryption techniques to secure both visual and non-visual data before transmitting the data to the data collector, safeguarding sensitive information.**Data Transmission:** UAVs collaboratively and securely transmit encrypted data to the data collector, contributing to the collective security framework.**Real-time Communication:** UAVs retain real-time communication links with one another and with the smart city control center, allowing for rapid information sharing and reaction to safety issues.**Autonomous Operation:** UAVs function independently, making decisions and adapting to changing situations without human interference, hence, improving monitoring and surveillance abilities.

### 3.2. Privacy-Preserving Ensemble

The privacy-preserving ensembles in the proposed model are crucial components suggested for securing sensitive data during intrusion detection and prevention processes. In this collaborative approach, the IDS can take advantage of the collective intelligence of UAVs while maintaining data privacy. The ensemble integrates machine learning models from various UAVs while maintaining the privacy of individual data points. A brief description of each technique follows.

#### 3.2.1. Federated Learning

Machine learning techniques such as federated learning enable models to be trained on different devices or nodes without transferring raw data. After combining all UAV models, data privacy is protected because the actual data are not shared after they are combined. Each UAV stores its own local data and uses these to train its own model.

Suppose $D=D_{1},D_{2},\ldots ,D_{n}$ represents local datasets from *n* UAVs, where θ represents the global model parameters, and F(θ) represents the global loss function. Based on its local data $D_{i}$, each UAV *i* trains its local model to minimize its own local loss function $f_{i}\left(\theta \right)$. The local models are then sent to a central server where they are combined to update the global model parameters as follows:(1)$$\theta_{\mathrm{global}}\leftarrow \mathrm{Aggregate}\left(\theta_{\mathrm{global}},\theta_{1},\theta_{2},\ldots ,\theta_{n}\right)$$

As part of the aggregation process, a weighted average of local model parameters is calculated according to the size of each UAV’s dataset. Upon updating the global model, the UAVs are then sent back to iterate until convergence is achieved.

Algorithm 1 describes clearly how our proposed federated learning model works.
**Algorithm 1** Federated Learning Algorithm
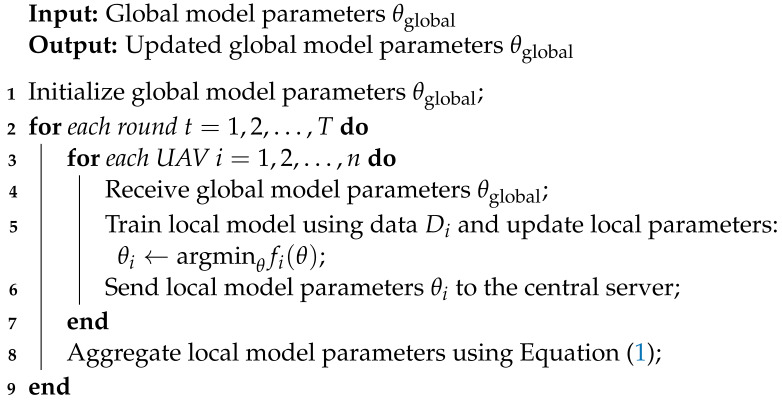


Several UAVs are involved in the algorithm, which operates in rounds. Global model parameters are initialized and shared with UAVs for local model training. UAVs update the global model using their local models, protecting data privacy. As UAV privacy is protected, federated learning improves intrusion detection and prevention.

#### 3.2.2. Differential Privacy

With our proposed model, we implement a fundamental privacy-preserving technique called differential privacy that protects individual data points while allowing accurate analysis of aggregate data. Differential privacy prevents sensitive information from being extracted about particular individuals or entities by adding carefully calibrated noise to the data. It can be defined as follows:

Given two neighboring datasets $D_{1}$ and $D_{2}$ that differ in only one data point, and a privacy parameter ε (epsilon), a randomized algorithm *M* provides ε-differential privacy if for all possible output sets *S*:(2)$$\mathrm{Pr}\left[M\left(D_{1}\right)\in S\right]\leq e^{\varepsilon }\cdot \mathrm{Pr}\left[M\left(D_{2}\right)\in S\right]$$

As part of our proposed model, each UAV data point is perturbed with differential privacy noise before it is sent to the central server for analysis. In addition to preserving the overall utility of the data for analyzing trends and patterns, this noise ensures that the statistical properties of the data remain intact.

Our proposed differential privacy model involves the following steps:(3)$$\mathrm{Sensitivity}=\max_{\mathrm{neighboring}\;\mathrm{datasets}\;D_{i},D_{i}^{\prime }}{∥x-x^{\prime }∥}_{1}$$ For each data point $x\in D_{i}$:(4)$$\Delta x=\max_{\mathrm{neighboring}\;\mathrm{datasets}\;D_{i},D_{i}^{\prime }}{∥x-x^{\prime }∥}_{1}$$ Apply Laplace noise to each data point:(5)$$x^{\prime }=x+\mathrm{Lap}\left(0,\Delta x/\varepsilon \right)$$

Our proposed differential privacy model is explained in Algorithm 2.
**Algorithm 2** Differential Privacy Algorithm
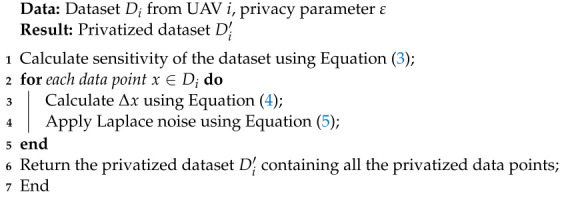


With differential privacy, UAV data privacy is preserved by adding Laplace noise before transmission. Based on the privacy parameter ε, sensitivity is calculated to determine the noise magnitude. As a result, the original data are protected while allowing accurate analysis at the central server.

#### 3.2.3. Secure Multi-Party Computation (SMC)

SMC is a fundamental cryptographic technique used in our proposed model to ensure data privacy and security. The IDS we provide enables multiple UAVs to collaborate on encrypted data in one solution.

As part of the SMC process, data are encrypted before they are shared with other participants. UAVs hold secret keys that encrypt their data before sharing these data as encrypted inputs with other UAVs. Since the actual computations take place on encrypted data, no UAV is able to access plain text data held by other participants.

Let ${\mathrm{UAV}}_{i}$ represent a UAV, and $D_{i}$ be its encrypted dataset. Each UAV computes a function $f_{i}$ on its encrypted data:(6)$$E_{i}=\mathrm{Encrypt}\left(D_{i},{\mathrm{key}}_{i}\right)$$
where Encrypt(·) denotes the encryption function using the UAV’s secret key ${\mathrm{key}}_{i}$, and $E_{i}$ is the encrypted dataset shared with other UAVs.

After receiving the encrypted inputs $E_{i}$ from all UAVs, the collaborative computation is performed using secure cryptographic protocols. The result is an encrypted output that contains the collective insights from all UAVs without exposing any individual UAV’s data.

The secure computation can be represented as:(7)$$E_{\mathrm{out}}=\mathrm{Compute}\left(E_{1},E_{2},\ldots ,E_{n}\right)$$
where Compute(·) denotes the secure computation function, and $E_{\mathrm{out}}$ is the encrypted output shared with all UAVs.

By utilizing advanced cryptographic protocols like secure multi-party computation, the proposed model enables UAVs to collaboratively train machine learning models and collectively detect anomalies without the need to disclose raw data to other entities. This privacy-preserving mechanism enhances the overall security of UAV operations in a smart city environment and strengthens the resilience of the IDS against potential attacks.

SMC plays a pivotal role in the collective decision-making process, allowing UAVs to share insights from their individual data sets without compromising data privacy. This collaborative approach facilitates the detection of sophisticated security threats that may span across multiple UAVs and ensures that the system operates effectively in real-time, adapting to evolving security challenges within the Internet of Unmanned Aerial Vehicles (IoUV) ecosystem.

Algorithm 3 clearly describes the functionality of SCM in our proposed model.
**Algorithm 3** Secure Multi-party Computation Algorithm
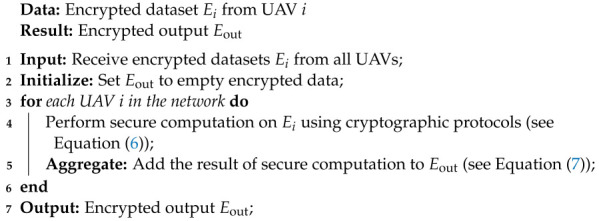


The secure multi-party computation algorithm facilitates data exchange and collaborative computations using encrypted datasets, producing the encrypted collective output $E_{\mathrm{out}}$. This method enables collective learning without revealing individual UAVs’ data, ensuring data privacy within the privacy-preserving ensemble. Real-time updates of the ensemble models with new data further enhance the IDS’s effectiveness in dynamic environments, enabling robust threat detection and prevention.

### 3.3. Intrusion Detection System (IDS)

As part of our proposed model for detecting and preventing intrusions in the Internet of UAVs ecosystem, the IDS is a critical component. Within the UAV network, it efficiently identifies threats and abnormalities.

#### 3.3.1. Data Collector

The data collector is a critical component of our proposed IDS model, responsible for efficiently collecting encrypted data from the privacy-preserving ensemble. This ensemble, consisting of federated learning, differential privacy, and secure multi-party computation techniques, securely aggregates data from multiple UAVs while preserving individual data privacy. Using advanced cryptographic protocols, the data collector receives the encrypted dataset $E_{i}$ from the privacy-preserving ensemble, where $E_{i}$ represents the collective data from multiple UAVs. Each UAV’s data are encrypted using its respective secret key $key_{i}$ before transmission to the data collector. The encryption process can be represented as:(8)$$E_{i}=\mathrm{Encrypt}\left(D_{i},key_{i}\right)$$
where Encrypt(·) denotes the encryption function using the UAV’s secret key, and $D_{i}$ is the dataset collectively aggregated from multiple UAVs.

Algorithm 4 outlines the steps performed by the data collector:
**Algorithm 4** Data Collection Algorithm
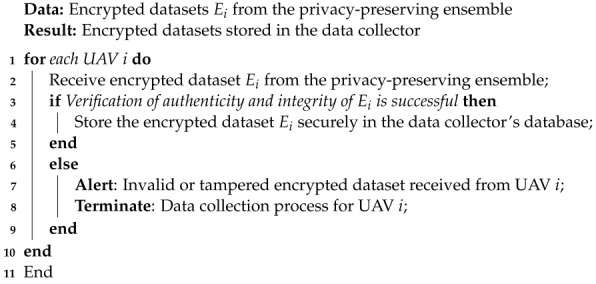


This algorithm securely receives encrypted datasets $E_{i}$ from the privacy-preserving ensemble, aggregating data from multiple UAVs. It then verifies the authenticity and integrity of the data before securely storing the data in its database. This collaborative approach empowers the intrusion detection system (IDS) to efficiently analyze data from multiple UAV sources while preserving data privacy, enhancing the security and responsiveness of UAV operations within the smart-city environment.

#### 3.3.2. Intrusion Detection Engine

The intrusion detection engine is a critical component within the proposed model responsible for efficiently identifying potential security threats and abnormal activities within the UAV network. It plays a key role in analyzing the collected and preprocessed data to detect any suspicious patterns or behaviors that may indicate intrusion attempts or anomalous activities. The intrusion detection engine is composed of two main stages: data preprocessing and anomaly detection and classification.
**Data Preprocessing:** Before feeding the data into the anomaly detection and classification stage, the data preprocessing stage performs essential tasks to ensure the data are in a suitable format for analysis. Data normalization, feature extraction, and data transformation are all part of this stage.**Data normalization:** Data normalization is an important step in data preprocessing that ensures all features are of equal importance and scale. Normalization techniques vary, but one popular technique is min-max scaling. The data are scaled to a fixed range, typically between 0 and 1, using the following formula:
(9)$$X_{\mathrm{normalized}}=\frac{X-X_{\min}}{X_{\max}-X_{\min}}$$ where *X* represents the original data, $X_{\min}$ represents the dataset’s minimum value, and $X_{\max}$ represents the dataset’s maximum value.

**Feature extraction:** Feature extraction entails extracting the most relevant features from the collected data. Principal component analysis (PCA) is a common feature extraction technique. PCA converts the data into a new coordinate system in which the new features, known as principal components, are orthogonal to each other and capture the most variance in the data.

In this step, we will need to calculate the covariance matrix C, which can be calculated as follows:(10)$$C=\frac{1}{n-1}\sum_{i=1}^{n}\left(x_{i}-\bar{x}\right){\left(x_{i}-\bar{x}\right)}^{T}$$
where *n* is the number of data points, $x_{i}$ represents each data point, and $\bar{x}$ is the mean vector of the data points. The term $\left(x_{i}-\bar{x}\right){\left(x_{i}-\bar{x}\right)}^{T}$ represents the outer product of the centered data point with itself. We will also need to compute the eigenvectors and eigenvalues of C, as follows:(11)C·V=V·Λ
where *C* represents the covariance matrix of the dataset, *V* represents a matrix containing the eigenvectors, and Λ is a diagonal matrix containing the eigenvalues.

After that, we have to compute the transformed data using the PCA equation. PCA is expressed as follows:(12)$$X_{\mathrm{transformed}}=X\cdot V$$
where *X* represents the original data matrix and *V* is the eigenvector matrix derived from the data covariance matrix. **Data transformation techniques:** These techniques such as t-distributed stochastic neighbor embedding (t-SNE) are used to further enhance the data representation, especially for visualization purposes. t-SNE is commonly used to reduce high-dimensional data to a lower-dimensional space while preserving the local structure of the data. Here, we compute the pairwise distances between data points in $X_{\mathrm{transformed}}$ as follows:
(13)$$d\left(x_{i},x_{j}\right)=\sqrt{\sum_{k=1}^{n}{\left(x_{ik}-x_{jk}\right)}^{2}}$$ where $d(x_{i},x_{j})$ represents the pairwise Euclidean distance between data points $x_{i}$ and $x_{j}$, *n* is the number of dimensions (features) in the data, and $x_{ik}$ and $x_{jk}$ are the *k*-th features of data points $x_{i}$ and $x_{j}$, respectively. t-SNE can be represented as follows:(14)$$P_{ij}=\frac{\exp\left(-\frac{|x_{i}-x_{j}{|}^{2}}{2\sigma_{i}^{2}}\right)}{\sum_{k\neq i}\exp\left(-\frac{|x_{i}-x_{k}{|}^{2}}{2\sigma_{i}^{2}}\right)}$$
(15)$$Q_{ij}=\frac{\left(1+\right|y_{i}-y_{j}{{|}^{2})}^{-1}}{\sum_{k\neq i}\left(1+\right|y_{i}-y_{k}{{|}^{2})}^{-1}}$$
(16)$$C=\sum_{i}\mathrm{KL}\left(P_{i}|Q_{i}\right)=\sum_{i}\sum_{j}P_{ij}\log\frac{P_{ij}}{Q_{ij}}$$
where $x_{i}$ and $x_{j}$ are data points in the original space, $y_{i}$ and $y_{j}$ are their corresponding points in the lower-dimensional space, $\sigma_{i}$ is the variance of the Gaussian distribution around $x_{i}$, $P_{ij}$ is the conditional probability that $x_{i}$ would pick $x_{j}$ as its neighbor, $Q_{ij}$ is the conditional probability that $y_{i}$ would pick $y_{j}$ as its neighbor, and $\mathrm{KL}(P_{i}|Q_{i})$ is the Kullback–Leibler divergence between $P_{i}$ and $Q_{i}$.

These data preprocessing steps, including normalization, feature extraction with PCA, and data transformation using t-SNE, are essential for improving the data representation and facilitating more effective anomaly detection and classification in the intrusion detection engine.

Algorithm 5 summarizes the data preprocessing stage.
**Algorithm 5** Data Preprocessing Algorithm
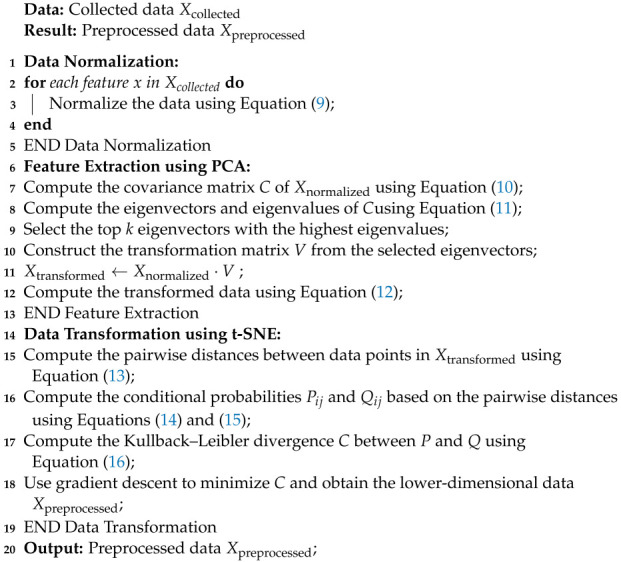


**Anomaly Detection and Classification:** In this stage, the preprocessed data are fed into advanced machine learning algorithms, such as the convolutional neural network-long short-term memory (CNN-LSTM) network, for anomaly detection and classification. The combination of CNN and LSTM allows the system to perform comprehensive anomaly detection and classification, enhancing the intrusion detection system’s ability to effectively identify and respond to security threats in the UAV network.**Convolutional Neural Network (CNN):** CNN is a powerful deep learning model commonly used for image and spatial data analysis. The CNN plays a critical role in anomaly detection in the proposed intrusion detection system for UAVs by learning spatial patterns and features from preprocessed data. The CNN architecture is made up of several layers, such as convolutional layers, pooling layers, and fully connected layers.

The convolution operation is the essential building component of a CNN. It entails applying filters (kernels) to input data in order to derive feature maps. In the output feature map, the convolution process for a specific pixel can be expressed as:(17)$$y_{ij}=\sum_{m}\sum_{n}w_{mn}\cdot x_{\left(i+m)(j+n\right)}$$
where $y_{ij}$ i represents the output feature map, $w_{mn}$ is the filter (kernel) value at position (m,n), and $x_{\left(i+m)(j+n\right)}$ represents the input data at position (i+m,j+n).

The activation function adds nonlinearity to the CNN, allowing it to simulate complicated data relationships. ReLU (rectified linear unit) and sigmoid are two common activation functions. The ReLU activation function is defined as follows:(18)f(x)=max(0,x)

Pooling layers reduce the spatial dimensions of the feature maps, lowering computational cost and preventing overfitting. The MaxPooling operation picks the largest value inside a particular window, while the Average Pooling operation computes the average value. The MaxPooling procedure can be represented as follows:(19)$$y_{ij}=\max\left(x_{\left(i+m)(j+n\right)}\right)$$
where $y_{ij}$ represents the pooled output, and $x_{\left(i+m)(j+n\right)}$ represents the input data in the pooling window.

Algorithm 6 summarizes the functionality of CNN in our proposed model.
**Algorithm 6** Convolutional Neural Network (CNN) Algorithm
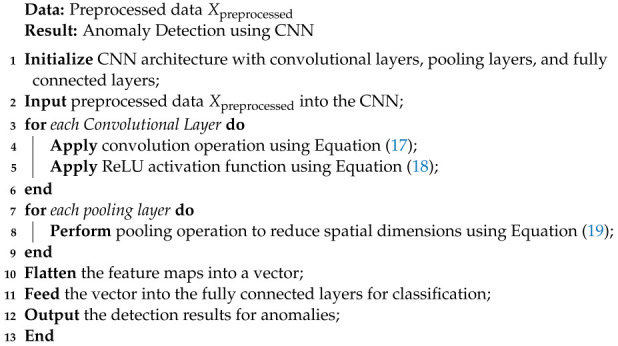


The CNN is largely in charge of recognizing and differentiating anomalies from routine UAV operations. It is particularly good at learning spatial features and patterns from preprocessed data. As a result, by evaluating geographical information, it may detect unexpected or abnormal patterns in data, indicating potential security threats or breaches.

**Long Short-Term Memory (LSTM):** The LSTM is a recurrent neural network (RNN) variant designed to handle sequential data, making it well-suited for time-series analysis and capturing temporal dependencies. It is important in the context of our proposed IDS as it takes detected anomalies from the convolutional neural network (CNN) and classifies them into specific categories or types based on their temporal characteristics. The LSTM is composed of memory cells that accumulate information over time, allowing it to remember and learn patterns that span multiple time steps. Its unique ability to retain long-term dependencies enables it to recognize complex temporal patterns in the sequence of detected anomalies, facilitating accurate anomaly classification.

The LSTM computationally processes sequential data by employing a set of gating mechanisms that regulate the flow of information. The input gate, forget gate, and output gate are examples of these mechanisms. The LSTM’s cell state stores information over time, while the gates determine how much information is retained or discarded at each time step. The following are the LSTM equations:


**Input Gate:**

(20)
$$i_{t}=\sigma \left(W_{xi}\cdot x_{t}+W_{hi}\cdot h_{t-1}+b_{i}\right)$$




**Forget Gate:**

(21)
$$f_{t}=\sigma \left(W_{xf}\cdot x_{t}+W_{hf}\cdot h_{t-1}+b_{f}\right)$$




**Output Gate:**

(22)
$$o_{t}=\sigma \left(W_{xo}\cdot x_{t}+W_{ho}\cdot h_{t-1}+b_{o}\right)$$




**Candidate Cell State:**

(23)
$$\tilde{C}t=\tanh\left(Wxc\cdot x_{t}+W_{hc}\cdot h_{t-1}+b_{c}\right)$$




**Cell State Update:**

(24)
$$C_{t}=f_{t}⊙C_{t-1}+i_{t}⊙{\tilde{C}}_{t}$$



**Hidden State Update:**(25)$$h_{t}=o_{t}⊙\tanh\left(C_{t}\right)$$ where $x_{t}$ is the input at time step *t*, $h_{t}$ is the hidden state at time step *t*, $C_{t}$ is the cell state at time step *t*, $i_{t}$, $f_{t}$, and $o_{t}$ are the input, forget, and output gates’ activations at time step *t*, respectively, ${\tilde{C}}_{t}$ is the candidate cell state at time step *t*, *W* and *b* are weight matrices and bias vectors, and σ is the sigmoid activation function, and ⊙ represents element-wise multiplication. Algorithm 7 summarizes the functionality of LSTM in our proposed model.
**Algorithm 7** Long Short-Term Memory (LSTM) Algorithm
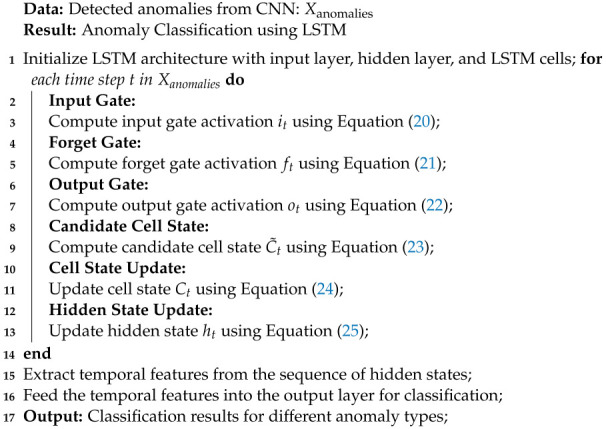


LSTM is chosen in our proposed model for classifying intrusions due to its exceptional ability to capture temporal dependencies within sequential data. Its unique architecture analyzes the temporal dynamics of anomalies over time, recognizing recurrent patterns and sequential behaviors crucial for accurate anomaly classification. By leveraging LSTM, our IDS effectively identifies and categorizes diverse intrusion scenarios, enhancing UAV operations’ security in a smart-city environment. Categorizing intrusions offers benefits such as improved response strategies, focused investigation, enhanced situational awareness, adaptive anomaly detection, and simplified forensic analysis, enhancing overall system reliability and safety in a dynamic smart-city environment.

Figure 2 illustrates the hybrid CNN-LSTM Model for intrusion detection and classification in our proposed SP-IoUAV model.

#### 3.3.3. Real-Time Decision Mechanism

The real-time decision mechanism, a critical component of our proposed model, comprises blacklist management and real-time alert provision.

**Blacklist Management:** Blacklist management is a crucial part of the real-time decision mechanism in our proposed model. It involves maintaining and updating a blacklist that contains identified malicious entities, such as unauthorized intruders or suspicious activities. The primary goal of blacklist management is to prevent future interactions with known threats and enhance the security of the unmanned aerial vehicle (UAV) network.

We can represent the blacklist as follows:(26)$$\mathrm{Blacklist}=e_{1},e_{2},\ldots ,e_{n}$$
where $e_{i}$ represents the *i*-th entity or intrusion identified as a threat.

Algorithm 8 summarizes the blacklisting functionality in our proposed model.

The blacklist management algorithm first checks if a new intrusion, $e_{\mathrm{new}}$, is already present in the current blacklist. If not, it adds the new intrusion to the blacklist. If $e_{\mathrm{new}}$ is already present, the algorithm updates its entry with any new information. Finally, the algorithm returns the updated blacklist, ${\mathrm{Blacklist}}_{\mathrm{updated}}$, which is then used to prevent any future interactions with known threats.

By continuously updating the blacklist with newly identified threats, the blacklist management component ensures that the UAV network remains protected from previously encountered malicious entities, significantly reducing the risk of security breaches and enhancing the overall security posture of the system.
**Algorithm 8** Blacklist Management Algorithm
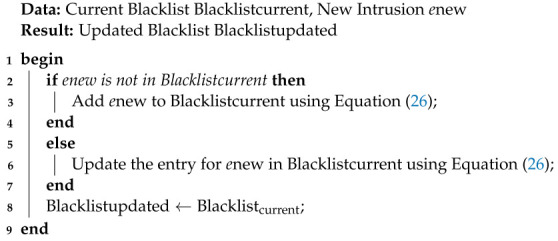


**Real-Time Alert Provision:** In our proposed model, real-time alert provision serves as a crucial mechanism for promptly notifying key stakeholders about detected security threats. Specifically, the mechanism sends alerts to the smart city control center, the privacy-preserving ensemble, and individual UAVs.

When it comes to UAVs, real-time alert provision offers flexibility in notification methods. Depending on the nature and severity of the detected intrusion, UAV operators can receive notifications either on a global level, where all UAVs are informed collectively, or on an individual level, where each UAV is notified separately. This adaptability allows UAV operators to tailor their response strategies based on the specific context of the intrusion.

By notifying the smart city control center, the privacy-preserving ensemble, and UAVs in a timely manner, our proposed model ensures a coordinated and swift response to security threats. This proactive approach enhances the overall security and resilience of UAV operations in the dynamic smart-city environment, facilitating effective threat mitigation and safeguarding critical assets and data.

Algorithm 9 summarizes the real-time alert provision mechanism.
**Algorithm 9** Real-Time Alert Provision Algorithm
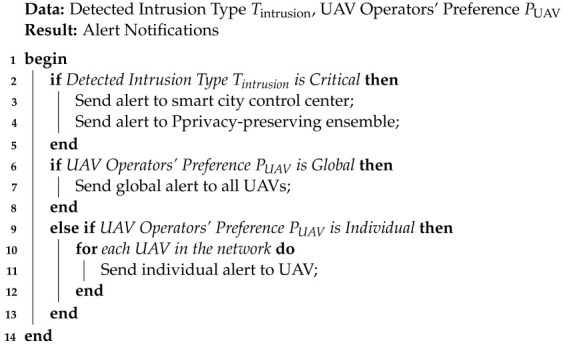


In the real-time alert provision algorithm, the variables $T_{\mathrm{intrusion}}$ and $P_{\mathrm{UAV}}$ hold crucial roles. $T_{\mathrm{intrusion}}$ represents the type of detected intrusion, while $P_{\mathrm{UAV}}$ captures the UAV operators’ preferences for alert notifications, whether they prefer global alerts or individual alerts. The algorithm begins by assessing the severity of the detected intrusion. For critical intrusions, alerts are dispatched to the smart city control center, the privacy-preserving ensemble, and UAVs based on their specified preference. In cases where the intrusion is not deemed critical, the algorithm sends alerts exclusively to UAVs, again adhering to their preferred notification method. By employing this approach, the algorithm ensures that relevant entities receive timely notifications tailored to the nature of the intrusion and the specific preferences of UAV operators.

#### 3.3.4. Intrusion Detection Database

The database is a crucial component in our proposed model, serving as a central repository for securely storing and managing critical information related to the intrusion detection system (IDS). It plays a pivotal role in efficiently storing preprocessed data collected by the data collector from various sources within the UAV network. These data are critical for the intrusion detection engine’s real-time decision-making, anomaly detection, and classification, which employs powerful convolutional neural network-long short-term memory (CNN-LSTM) algorithms.

In addition, the database maintains a complete record of all identified intrusions and their types, providing valuable historical references for analysis and auditing. These historical data allow the IDS to fine-tune its anomaly detection algorithms and response tactics, improving the overall efficacy and resilience of the intrusion detection and prevention system. Moreover, the database is critical in blacklist management, which protects the UAV network by preventing future contacts with known dangerous entities or sources of intrusion attempts.

**Database Protection**: Our proposed model employs a strong multi-factor authentication (MFA) mechanism to ensure the highest level of security for sensitive data and secure system access. The MFA system combines three types of authentication: traditional username and password, and advanced facial recognition technology. In the first layer of authentication, users must provide a unique username and a strong, complex password during the login process. By preventing unauthorized access and protecting against brute-force attacks, this traditional method adds a fundamental level of security. A username and password combination serves as an important security barrier, allowing only authorized users to access the system.

Facial recognition is used as an additional authentication layer in our model to increase security. During the login process, the system uses a camera or an image sensor to capture the user’s facial features. These facial features are then compared to the stored biometric data to confirm the user’s identity. Facial recognition adds a biometric factor to the authentication process, making it extremely difficult for unauthorized users to impersonate legitimate users. This advanced layer of security significantly strengthens the system’s defense against unauthorized access attempts.

By integrating these three authentication factors, multi-factor authentication ensures a robust and secure access control system. Even if an attacker manages to obtain the username and password through phishing or other means, they would still need to pass the facial recognition step, ensuring that only authorized users gain access to the system. As a result, our intrusion detection system (IDS) for unmanned aerial vehicles (UAVs) in the smart-city environment remains protected from potential security breaches and data breaches, upholding data privacy and system integrity.

Algorithm 10 summarizes the database accessibility.
**Algorithm 10** Database Accessibility Algorithm with MFA
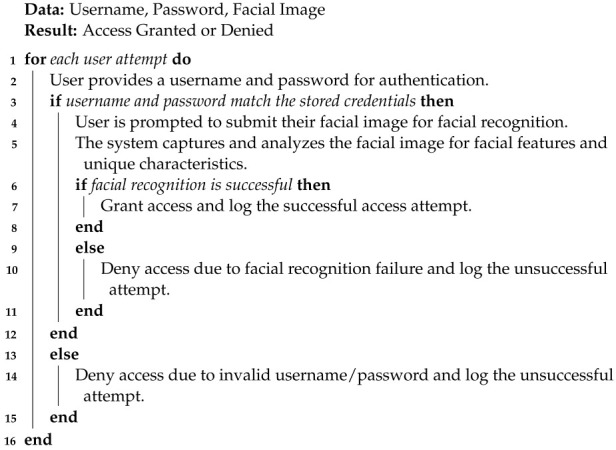


### 3.4. Smart City Control Center

The smart city control center is a critical component in our proposed model, serving as the central command hub for overseeing and managing the UAV network’s security and operations within the smart city environment. It gathers real-time data from the intrusion detection system (IDS) and other surveillance systems deployed across the city. Utilizing advanced data analytics and AI algorithms, it processes and analyzes incoming data to identify potential security threats and anomalies in the UAV network. Prompt response to detected intrusions and appropriate actions to mitigate security risks are facilitated by leveraging IDS outputs.

Moreover, the smart city control center coordinates UAV responses during emergencies, disaster management, and citywide events. It dynamically adjusts UAV flight paths, allocates resources, and communicates with operators to ensure efficient UAV operations. To enhance security and access control, the center implements multi-factor authentication (MFA) requiring username, password, and facial recognition for authorized personnel to access the system. This multi-layered security approach ensures that only authorized individuals access critical data and functionalities, safeguarding the UAV network’s integrity and sensitive information.

With its real-time situational awareness, intelligent decision-making capabilities, and robust security measures, the smart city control center plays a pivotal role in enabling safe, efficient, and secure UAV operations in the smart city environment. Its centralized management and control contribute to the overall success of our proposed model, ensuring privacy-preserving, secure, and resilient UAV operations. In the next section, we present “Experimental Findings and Analysis”, showcasing results and performance evaluation.

## 4. Experimental Findings and Analysis

In this section, we present a comprehensive evaluation of our proposed model’s performance. This section is divided into five subsections, namely, “System Requirements”, where we outline the hardware and software configurations used for experimentation, “Dataset Description”, which provides insights into the dataset employed in the evaluation, “Simulation Setup”, where we presents simulation parameters used in this research work, “Comparative Analysis”, where we compare the results of our proposed model with existing intrusion detection systems, and “Security Analysis of the Proposed Model”, where we assess the model’s robustness against various security threats. The experimental findings and analysis aim to shed light on the effectiveness and reliability of our proposed intrusion detection system within a smart-city environment.

### 4.1. System Requirements

The system requirements subsection details the essential hardware and software prerequisites for the effective implementation and operation of our system. Meeting these specific requirements, as presented in Table 3, is crucial to ensure the successful installation and smooth execution of the system.

### 4.2. Dataset Description

Given the absence of a standardized intrusion dataset specifically tailored to the UAV domain, our proposed model utilized the state-of-the-art Canadian Institute of Cybersecurity dataset, CICIDS2017 [36], for both training and testing. Specifically, 80% of the dataset was allocated for training, allowing the model to learn intricate patterns and behaviors. The remaining 20% was reserved for testing, enabling the assessment of the model’s performance on previously unseen data, as illustrated in Table 4.

This dataset comprehensively covers various normal and diverse contemporary attacks, such as Brute Force FTP, Brute Force SSH, DoS, Web Attack, Infiltration, Botnet, and DDoS. The decision to select CICIDS2017 over other options was driven by its inclusion of complex and up-to-date network attack types not found in alternative datasets. Moreover, CICIDS2017 contains a mixture of benign and the latest common attacks, closely resembling real-world data in the form of PCAPs. Additionally, it includes labeled flows with essential details, such as timestamp, source and destination IPs, source and destination ports, protocols, and attack information, stored in CSV files, facilitating network traffic analysis using CICFlowMeter.

### 4.3. Simulation Setup

In the simulation setup subsection, we provide specific details regarding the parameters and configurations used to evaluate the performance of the SP-IoUAV model. For our simulation environment, we utilized the ns3 simulator, which allows us to recreate real-world scenarios to assess the effectiveness of our proposed model in detecting and preventing intrusions in UAV networks. In our simulation, we considered a total of 50 UAV nodes, each equipped with a communication range of 200 m to enable effective communication within the network. To replicate the dynamic movement patterns of UAVs in a smart-city environment, we implemented realistic mobility models, such as Random Waypoint and Manhattan. For simulating real-world traffic scenarios, we integrated the CIC-IDS2017 dataset through the OMNeT++ framework, providing comprehensive smart-city traffic patterns for evaluation. Additionally, we introduced various intrusion scenarios, including data injection attacks and unauthorized access attempts, to assess the SP-IoUAV model’s ability to detect and prevent different types of intrusions in UAV networks. By carefully defining these parameters and scenarios, we established a comprehensive and representative simulation environment, enabling rigorous testing of the SP-IoUAV model’s efficiency and robustness in safeguarding UAV operations within a smart-city setting.

### 4.4. Comparative Analysis

In this section, the proposed model’s performance is assessed by comparing it to several state-of-the-art methods based on the same dataset, such as FCL-SBL [31], RF-RSCV [30], and RBFNNs [27], considering parameters such as accuracy, precision, recall and *F*-Score. All performance results are based on the confusion matrix shown in Figure 3.

In machine learning, a confusion matrix is a table used to assess the performance of a classification model. It compares the predicted and true labels of the data and displays the number of true positives, true negatives, false positives, and false negatives. True positives are correctly identified positive cases, true negatives are correctly identified negative cases, false positives are negative cases that were incorrectly identified as positive, and false negatives are positive cases that were incorrectly identified as negative.

The confusion matrix is used to calculate metrics such as accuracy, precision, recall, and F1-score, which provide valuable insights into the model’s effectiveness in distinguishing between classes and identifying areas for improvement.

#### 4.4.1. Accuracy

Accuracy is a performance metric used to measure the overall correctness of a predictive model. It represents the proportion of correct predictions (both true positives and true negatives) made by the model out of all instances in the dataset. Its formula is as follows:(27)$$\mathrm{Accuracy}=\frac{TP+TN}{TP+TN+FP+FN}\times 100$$
where *TP* stands for true positives, *TN* stands for true negatives, *FP* stands for false positives, and *FN* stands for false negatives. Figure 4 clearly illustrates the accuracy of our proposed model.

The proposed SP-IoUAV model presents the highest accuracy of 99.98% compared to 95.5%, 99.89%, and 99.84% of the existing approaches, respectively. The reason behind this highest accuracy lies in its effective utilization of a combination of CNN-LSTM deep learning algorithms for anomaly detection and classification.

#### 4.4.2. Precision

Precision, in the context of classification models, is a performance metric that measures the proportion of true positive predictions among all positive predictions made by the model. It helps to assess the model’s ability to avoid false positive errors. It is calculated as follows:(28)$$Precision=\frac{TP}{TP+FP}$$

Figure 5 clearly illustrates the precision of our proposed model.

As illustrated in Figure 5, our proposed model presents the highest precision of 99.93% compared to 94.8%, 99.89% and 99.76% of the existing approaches, respectively. This is due to the robust data preprocessing techniques employed in our model, which includes data normalization, feature extraction, and data transformation. These preprocessing steps enhance the quality and relevance of the input data, resulting in better representation and separation of the positive samples from the negative ones during the training phase. As a consequence, the model achieves a higher proportion of true positive predictions among all positive predictions, minimizing false positive errors and leading to the improved precision observed in our results.

#### 4.4.3. Recall

Recall, also known as true positive rate or sensitivity, is a metric used in classification tasks to measure the ability of a model to correctly identify positive instances (i.e., the proportion of actual positive samples that were correctly predicted as positive by the model). The formula for recall is given as:(29)$$\mathrm{Recall}=\frac{TP}{TP+FN}$$

Figure 6 clearly illustrates the recall of our proposed SP-IoUAV Model.

As illustrated in Figure 6, our proposed model also presents the highest recall of 99.92% compared to 94.3%, 99.89% and 99.68% of the existing approaches, respectively. This is due to its ability to effectively detect and capture a larger proportion of true positive instances (intrusions) in the dataset. The incorporation of sophisticated deep learning algorithms, such as CNN-LSTM, and the utilization of a comprehensive data preprocessing approach that includes data normalization, feature extraction, and data transformation techniques contribute to the model’s exceptional performance in accurately identifying and recalling instances of intrusions, even in complex and dynamic UAV environments.

#### 4.4.4. *F*-Score

The *F*-Score, also known as the F1-score, is a performance metric used to evaluate the balance between precision and recall of a classification model. It considers both false positives and false negatives, making it a useful measure when dealing with imbalanced datasets. The formula for *F*-Score is given by:(30)$$F-\mathrm{Score}=\frac{2\times \mathrm{Precision}\times \mathrm{Recall}}{\mathrm{Precision}+\mathrm{Recall}}$$

Figure 7 clearly illustrates *F*-Score of our proposed model.

Our SP-IoUAV model has the highest *F*-Score of 99.92%, compared to 94.3%, 99.89%, and 99.68% of the existing approaches. This is due to the synergistic combination of precision and recall achieved by integrating CNN-LSTM deep learning algorithms for anomaly detection and classification. This combination enables our model to detect and classify intrusions with high accuracy while minimizing false positives and false negatives. Furthermore, data preprocessing techniques such as data normalization, feature extraction, and data transformation improve the model’s ability to handle complex and dynamic UAV network data, contributing to its superior *F*-Score performance.

Our proposed SP-IoUAV model outperformed previous approaches in terms of accuracy, precision, recall, and *F*-Score in this comparative analysis. Its exceptional performance was enabled by the integration of CNN-LSTM deep learning algorithms, data preprocessing, and a secure ensemble of federated learning, differential privacy, and secure multi-party computation. The real-time decision-making capability enhanced threat identification and response in dynamic environments. Overall, our SP-IoUAV model proved to be a robust and efficient solution for securing UAV-based applications in smart cities and other industries.

### 4.5. Security Analysis of the Proposed SP-IoUAV Model

In the security analysis of the SP-IoUAV model, we thoroughly evaluate its ability to counter a range of intrusions that could jeopardize UAV operations in a smart city. Our model effectively combats the following intrusions using advanced methodologies:**Distributed Denial of Service (DDoS) Attacks:** DDoS attacks pose a significant threat to UAV networks, aiming to disrupt services by flooding them with a high volume of malicious traffic. To counter this, our proposed model incorporates a sophisticated “intrusion Detection Engine”. Within this engine, the “Anomaly Detection and Classification” stage employs advanced machine learning algorithms, including the powerful CNN-LSTM. This stage plays a pivotal role in recognizing and classifying various anomalies, including DDoS attacks. It excels in identifying abnormal patterns within network traffic, making it a crucial component in safeguarding the UAV network within the dynamic environment of a smart city (refer to Section 3.3.2).**Man-in-the-Middle (MitM) Attacks:** MitM attacks attempt to intercept communication between UAVs and the smart-city infrastructure. Our model incorporates an “Intrusion Detection Engine” that actively monitors the network for any suspicious activities, providing an essential layer of defense against unauthorized access and ensuring ensuring data security (refer to Section 3.3.2).**Code Injection:** In the scenarios involving code injection, where malicious code is inserted into the UAV system to compromise its operations, the SP-IoUAV model employs a multi-step approach within the intrusion detection engine. This begins with robust preprocessing techniques, including crucial sub-steps like feature extraction, all designed to enhance the system’s ability to detect and prevent code injection attempts. Principal component analysis (PCA) plays a pivotal role in this process, enabling efficient feature extraction. These combined efforts ensure the integrity and seamless operation of the UAV system, even in the face of potential code injection threats (refer to Section 3.3.2).**Eavesdropping:** Eavesdropping involves unauthorized interception of data transmission. Our model utilizes secure communication channels and encryption techniques, along with t-distributed stochastic neighbor embedding (t-SNE) for data transformation, to safeguard data privacy and prevent eavesdropping attacks (refer to Section 3.3.2).**Spoofing:** Spoofing attacks involve impersonating legitimate UAVs or smart-city devices to gain unauthorized access. The SP-IoUAV model employs device authentication and verification mechanisms, along with min–max scaling for data normalization, to detect and thwart spoofing attempts (refer to Section 3.3.2).

In addition to these methodologies, our model also integrates multi-factor authentication (MFA) to protect the IDS database from various types of attacks, including brute-force attacks, credential stuffing, phishing, man-in-the-middle attacks, and account takeovers. MFA adds an extra layer of security by requiring users to provide multiple authentication factors, enhancing the overall security posture of our proposed SP-IoUAV model (refer to Section 3.3.4).

## 5. Conclusions and Future Research Directions

In this research, we proposed the SP-IoUAV model, an innovative approach addressing security and privacy concerns in UAVs within the Internet of UAVs ecosystem. By incorporating a hybrid privacy-preserving mechanism that combines federated learning, differential privacy, and secure multi-party computation, our model ensures robust data protection and enhances intrusion detection accuracy. The use of deep neural networks like CNN-LSTM in our intrusion detection engine enables precise and timely threat identification, enabling real-time decision-making in dynamic environments and bolstering UAV security in smart cities. The use of multi-factor authentication (MFA) further safeguards the IDS database. Our proposed model has been evaluated using the CIC-IDS2017 dataset. The results have shown its superiority over previous approaches, such as FCL-SBL, RF-RSCV, and RBFNNs. It achieved high levels of accuracy (99.98%), precision (99.93%), recall (99.92%), and *F*-Score (99.92%). While the SP-IoUAV model demonstrates promising results, it is important to acknowledge certain limitations. For instance, the model’s computational requirements may pose challenges in resource-constrained environments. Additionally, scalability and generalizability across diverse UAV mission scenarios require further investigation.

In future work, we aim to integrate blockchain technology for enhanced data security and transparency. Expanding and diversifying the experimental dataset will be pivotal in refining the SP-IoUAV model. Additionally, we will focus on real-time adaptability to dynamic network conditions, multi-modal data fusion for more effective intrusion detection, and improving energy efficiency on resource-constrained UAVs. These steps are essential for strengthening our model and advancing intelligent UAV systems to meet evolving security challenges in dynamic urban environments.

## Figures and Tables

**Figure 1 sensors-23-08077-f001:**
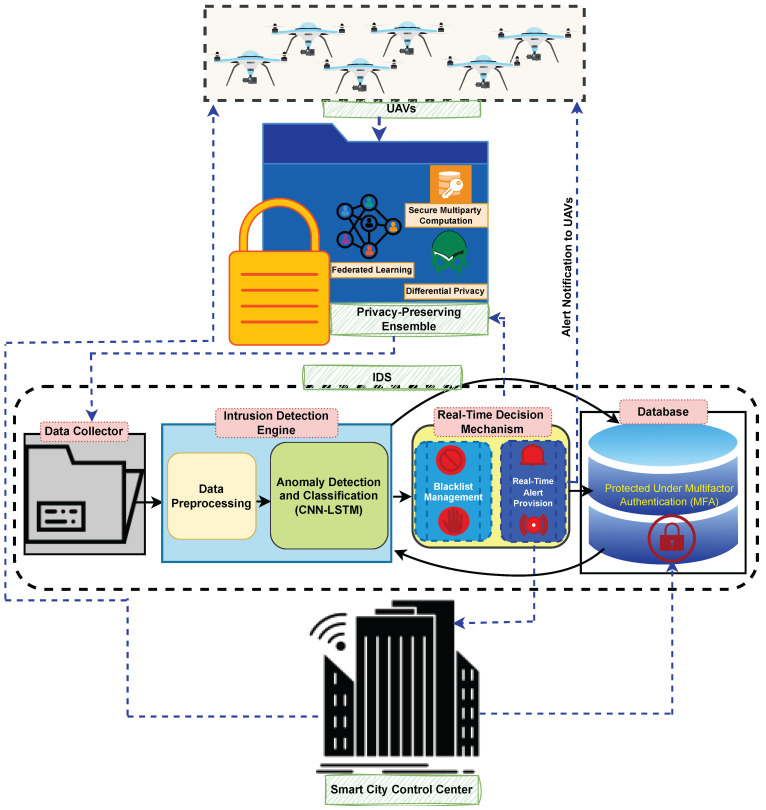
The Proposed SP-IoUAV Model.

**Figure 2 sensors-23-08077-f002:**
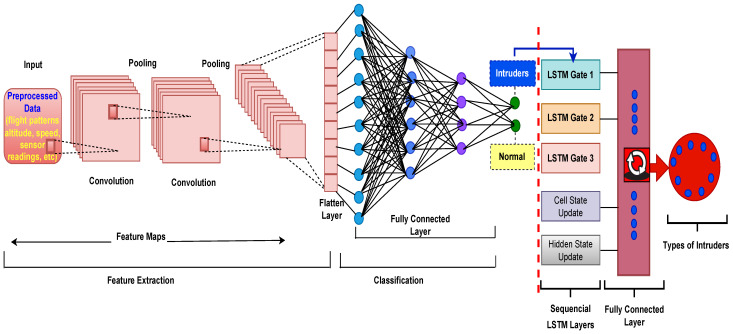
Hybrid CNN-LSTM Model for Intrusion Detection and Classification.

**Figure 3 sensors-23-08077-f003:**
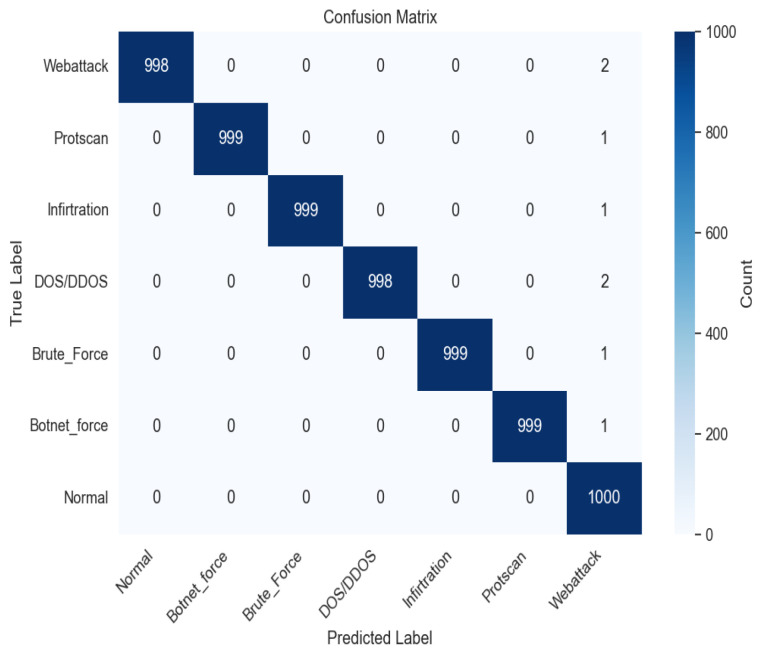
Confusion Matrix.

**Figure 4 sensors-23-08077-f004:**
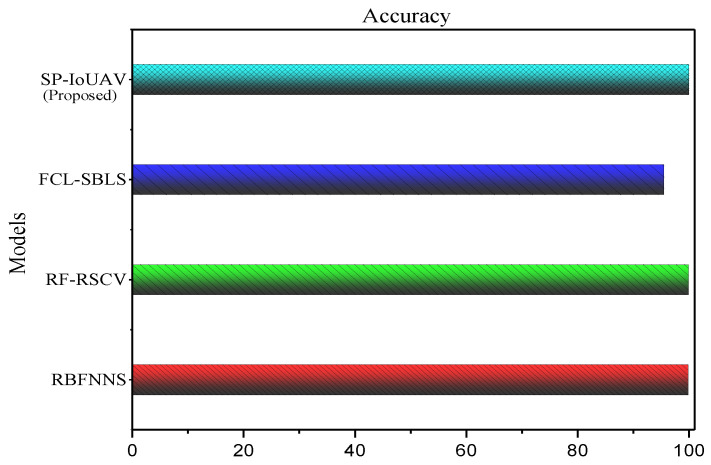
Accuracy of our proposed SP-IoUAV model.

**Figure 5 sensors-23-08077-f005:**
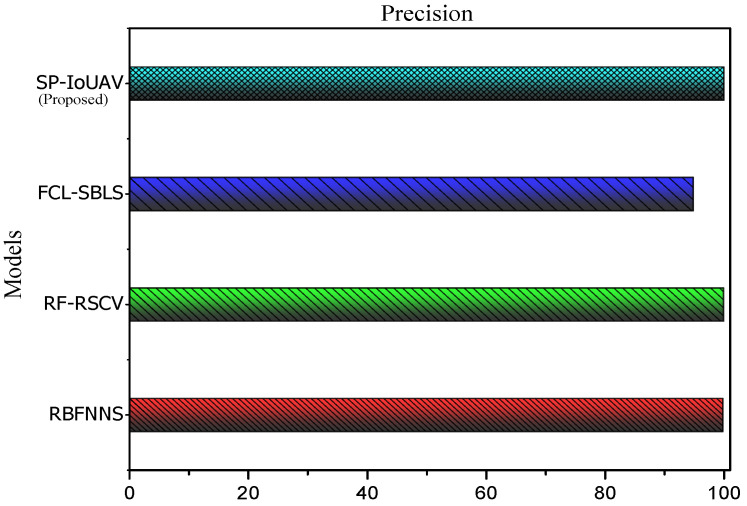
Precision of our proposed SP-IoUAV Model.

**Figure 6 sensors-23-08077-f006:**
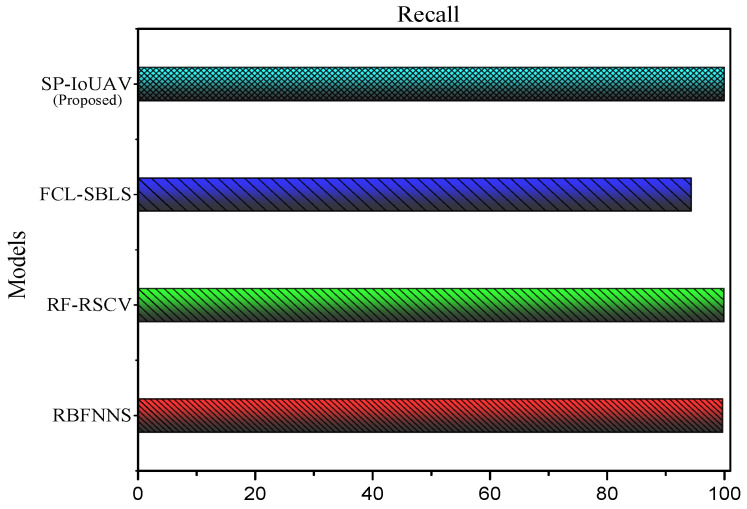
Recall of our proposed SP-IoUAV Model.

**Figure 7 sensors-23-08077-f007:**
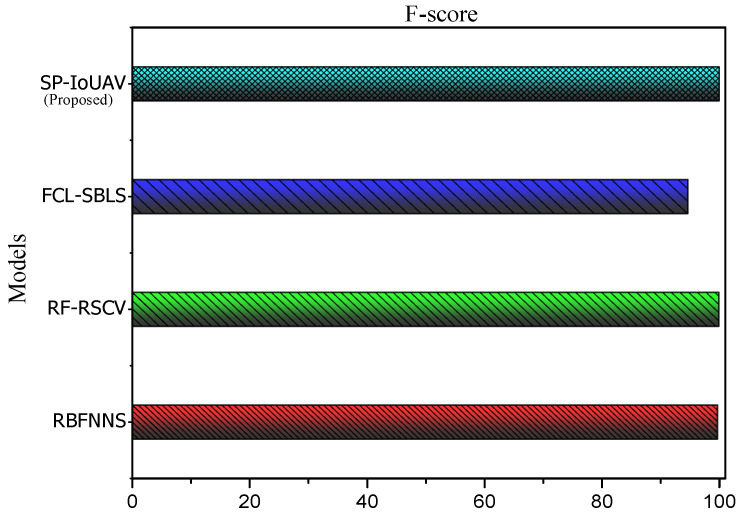
*F*-Score of our proposed SP-IoUAV Model.

**Table 1 sensors-23-08077-t001:** Related Works: Methods, Limitations, and Advantages.

Paper	Methods	Limitations	Advantages
[17]	Hybrid ML technique of Logistic Regression and Random Forest	Weak security measures	High accuracy (98.58%)
[18]	Enhanced K-vector perturbation and Random LNC (RLNC)	Insufficiency of sensitive data security	Explores UAV security, Decrypts locations
[23]	OC-SVM and K-Means++ Clustering	Unclear detection process	OC-SVM outperforms k-means
[24]	Neural Network Algorithm	Unclear methodology description	High detection, low false alarms
[25]	LSTM	Single NN less effective	Effective point anomaly detection
[26]	Hybrid ML (K-Nearest Neighbor (KNN), Naive Bayes (NB)), and blockchain	Unclear methodology description	Enhanced security, data reliability
[27]	Radial Basis Function Neural Network (RBFNN) and Blockchain	Need of BIIR Complements	High accuracy in attack identification
[28]	Supervised Learning (LightGBM and GANS), and Unsupervised Learning (One-class classifier and Federated Learning)	Efficiency evaluation gap	Wider attack coverage, privacy
[29]	CGAN, Blockchain, and LSTM	Need of ensemble approaches	Exceptional accuracy (>95%)
[30]	Mobile Edge Computing (MEG), Optimized Random Forest (RCSV)	Need of ensemble for improved accuracy	Efficient attack detection in various zones
[31]	Federated Learning, Deep Deterministic Policy Gradient (DDPG)	Ensemble technique needed	High accuracy, training efficiency
[32]	Random Forest	Ensemble technique needed for improved accuracy	Superiority in identifying lethal attacks
[33]	Supervised ML (J48, Classification via Regression, OneR and JRip)	Performance evaluation not systematic	High classification accuracy (>91%)
[34]	FSR protocol	Limited network scenarios	Efficient resource utilization, high delivery ratios
[35]	ELA and ANN Algorithms	Only rural image security focus	Authentic assessment of satellite images

**Table 2 sensors-23-08077-t002:** Notation Table.

Notation	Meaning
$D=\{D_{1},D_{2},\ldots ,D_{n}\}$	Local datasets from *n* UAVs
θ	Global model parameters
F(θ)	Global loss function
$D_{i}$	Local data of UAV *i*
$f_{i}\left(\theta \right)$	Local loss function of UAV *i*
$\theta_{i}$	Local model parameters of UAV *i*
$\theta_{\mathrm{global}}$	Updated global model parameters
$\mathrm{Aggregate}(\theta_{\mathrm{global}},\theta_{1},\theta_{2},\ldots ,\theta_{n})$	Aggregation process
ε	Privacy parameter (epsilon)
$E_{i}$	Encrypted dataset from UAV *i*
Δx	Sensitivity
Lap(0,Δx/ε)	Laplace noise
$E_{\mathrm{out}}$	Encrypted output
${\mathrm{UAV}}_{i}$	UAV *i*
$D_{i}^{\prime }$	Privatized dataset from UAV *i*
$E_{\mathrm{out}}$	Encrypted output
$X_{\mathrm{collected}}$	Collected data
$X_{\mathrm{normalized}}$	Normalized data
$X_{\mathrm{transformed}}$	Transformed data
$X_{\mathrm{preprocessed}}$	Preprocessed data
$y_{ij}$	Output feature map
$x_{\left(i+m)(j+n\right)}$	Input data at position (i+m,j+n)
$X_{\mathrm{anomalies}}$	Detected anomalies from CNN
$x_{t}$	Input at time step *t*
$h_{t}$	Hidden state at time step *t*
$C_{t}$	Cell state at time step *t*
$i_{t}$	Input gate activation at time step *t*
$f_{t}$	Forget gate activation at time step *t*
$o_{t}$	Output gate activation at time step *t*
${\tilde{C}}_{t}$	Candidate cell state at time step *t*
σ	Sigmoid activation function
⊙	Element-wise multiplication
$W_{xi},W_{xf},W_{xo},W_{xc}$	Weight matrices for input, forget, output, and candidate gate, respectively
$W_{hi},W_{hf},W_{ho},W_{hc}$	Weight matrices for input, forget, output, and candidate gate, respectively
$b_{i},b_{f},b_{o},b_{c}$	Bias vectors for input, forget, output, and candidate gate, respectively
$e_{i}$	*i*-th entity or intrusion identified as a threat
$T_{\mathrm{intrusion}}$	Detected Intrusion Type
$P_{\mathrm{UAV}}$	UAV Operators’ Preference

**Table 3 sensors-23-08077-t003:** System Requirements.

Specification Types	Name	Value
**Software**	Operating System	Microsoft Windows 11 Pro
	Programming Language	Python3
	Integrated Development Environment (IDE)	Visual Studio Code (VSCode)
	Development Framework	OpenCV
	Drawing and Visualization Tool	Draw.io
	Deep Learning Framework	TensorFlow
	Deep Learning Framework	PyTorch
	Simulation Tools	ns3, OMNeT++
	Data Analysis Tool	Origin Pro 9.0 64 bit
**Hardware**	Processor	Intel(R) Core(TM) i5-8400 CPU @ 2.80 GHz, 6 Core(s), 6 Logical Processor(s)
	RAM	24 GB
	Hard Disk	2 TB

**Table 4 sensors-23-08077-t004:** Training and Testing Data Distribution for CIC-IDS2017.

Classes	Train	Test
Web_Attack	1772	443
PortScan	126,160	31,540
Infiltration	29	7
DoS	310,268	77,567
Brute_Force	11,004	2751
Botnet	1552	388
Normal	1,808,288	452,072

## Data Availability

The data supporting reported results will be made available upon request.

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
