# Peer review of "Secure and Privacy-Preserving Intrusion Detection and Prevention in the Internet of Unmanned Aerial Vehicles"

_sensors, 2023, doi:10.3390/s23198077_

Round 1

Reviewer 1 Report

1) Is there any selection criteria for the dataset in your manuscript.

2) I would suggest that write at least 2-3 line at the ending paragraph in introduction section about the current limitations in the literature and your approach to address these limitations.

3) create a table at the end of the literature mentioning the limitations of each research paper with advantages. 

4) Make sure all tables, figures and mathematical formulas are referenced in the text.

5) write down limitations of your approach and possible future direction in the conclusion section. 

6) Following research papers can be cited in your manuscript:

a)10.1109/TGCN.2022.3157591

b)10.1109/ICRAI57502.2023.10089543

Reviewer 2 Report

What are the limitations of the existing works that motivated the current research?

Contents about research background in Abstract can be replaced by more details directly related to this paper, including research targets, approaches and experiments.

There are many grammar mistakes in the manuscript. Also, the punctuation and narration of this manuscript makes readers confusing. Please check.

Literature review section should have some critical analysis regarding innovative work. can be cited Preserving privacy of classified authentic satellite lane imagery using proxy re-encryption and UAV technologies

More experiments should be conducted to make the manuscript more persuasive. More datasets can be applied to test the algorithm 

Sections and subsections need to be appropriately arranged.

The quality of the figures can be improved more. Figures should be eye catching. It will enhance the interest of the reader. text in figures difficult to read here.

Discuss the future plans with respect to the research state of progress and its limitations.

 Moderate editing of English language required

Reviewer 3 Report

In abstract, there are few typo mistakes "a nd".

In abstract, problem statement and motivation statements are completely missing. 

Introduction section needs to be enhanced with the problem statement, motivation statement and related work. 

I suggest the authors to provide the detailed comparison table of existing approaches. The authors can take different parameters to compare them and provide a constructive view. 

I suggest the authors to used only once full form of words followed by their abbreviations then. 

Algorithms (1-4) included in the paper are very simple and straight forward. I suggest the authors to enhance them by adding them the detailed functionality and steps. 

Few typo mistakes are in section 3.3.2 such as (bullet point used for only one heading, the use of :)

The notations table is missing. I suggest the authors to first add the notation tables along with their meanings.

Security analysis section is very simple, which only included their definitions. 

Conclusion should also include the achieved significant results.

 Minor editing of English language required.

Round 2

Reviewer 2 Report

authors are addressed all comments given and now to do proofread to improve the readability.

Moderate editing of English language required